REGISTERED REPORT PROTOCOL

# Intravenous thrombolysis for acute central retinal artery occlusion: Protocol for a systematic review and individual participant data meta-analysis of randomized controlled trials

Brian Mac Grory[1,2,3,4*], Cécile Preterre[5], Pierre Lebranchu[5], Stephen James Ryan[6], Øystein Kalsnes Jørstad[7], Morten Carstens Moe[7], Johannes Tünnerhoff[8], Martin S. Spitzer[9], Carsten Grohmann[9], Oana M. Dumitrascu[9], Valérie Biousse[10,11], Benoit Guillon[4], Anne Hege Aamodt[6,12,13], Sven Poli[8,14], Matthew Schrag[15], The Central Retinal ArterY OcclusioN (CRAYON) Collaborators

1 Departments of Neurology, Duke University School of Medicine, Durham, North Carolina, United States of America, 2 Department of Ophthalmology, Duke University School of Medicine, Durham, North Carolina, United States of America, 3 Duke Clinical Research Institute, Durham, North Carolina, United States of America, 4 ORBIT Interdisciplinary Hub, Duke University, Durham, North Carolina, United States of America, 5 Departments of Neurology and Ophthalmology, CHU de Nantes - Laennec Nantes, France, 6 Department of Neurology, Oslo University Hospital, Oslo, Norway, 7 Department of Ophthalmology, Oslo University Hospital, and Faculty of Medicine, University of Oslo, Oslo, Norway, 8 Department of Neurology & Stroke, University of Tübingen, Tübingen, Germany, 9 Department of Ophthalmology, University Medical Center Hamburg Eppendorf, Hamburg, Germany, 10 Departments of Neurology and Ophthalmology, Mayo Clinic Alix School of Medicine, Phoenix, Arizona, United States of America, 11 Departments of Ophthalmology and Neurology, Emory University School of Medicine, Atlanta, Georgia, 12 Department of Neuroscience and Movement Science, The Norwegian University of Science and Technology, Trondheim, Norway, 13 Institute of Population Health, Faculty of Health and Life Sciences, University of Liverpool, Liverpool, United Kingdom, 14 Hertie Institute for Clinical Brain Research, University of Tübingen, Tübingen, Germany, 15 Department of Neurology, Vanderbilt University School of Medicine, Nashville, Tennessee, United States of America

* brian.macgrory@duke.edu

This is a Registered Report and may have an associated publication; please check the article page on the journal site for any related articles.

## Abstract

### Importance

Central retinal artery occlusion (CRAO) is a disabling subtype of acute ischemic stroke. It is not known whether intravenous (IV) thrombolysis delivered within 4.5 hours of time last known well (LKW) improves visual outcomes.

### Study design

Systematic review and individual participant data meta-analysis.

### Objective

The objective of this study is to determine whether IV thrombolysis improves visual outcomes among patients with acute non-arteritic CRAO when administered within 4.5 hours of time LKW compared with placebo, no IV thrombolysis, and/or anti-thrombotic therapy.

**Data availability statement:** The extracted data from studies may be made available upon request to the corresponding authors of each individual clinical trial, in accordance with any applicable regulations and data sharing policies.

**Funding:** No authors are employed by a commercial company. No commercial entities had any role in the design of the present study.

**Competing interests:** BMG is supported by the National Institutes of Health (K23HL161426, R03HL178686, and UG3NS138219), the American Heart Association (23MRFSCD1077188 & 25GLP1450119), Duke Bass Connections, the Duke Office of Physician-Scientist Development, and the Duke University Office of the Provost. CP has nothing to disclose. PL received speakers' honoraria/ consulting fees from Santhera, Novartis, Lissac-Optic2000, Alexion, and Amgen. SJR has nothing to disclose. ØKJ: Allergan (speaker, consultant), Bayer (speaker, consultant), Chiesi Farmaceutici (speaker), Roche (consultant), and SJJ Solutions (speaker, consultant, and royalties). MCM Allergan (consultant), Bayer (speaker, consultant), Roche (speaker, consultant), Apellis (consultant), Novartis (consultant) and SJJ Solutions (consultant, and royalties). JT has nothing to disclose. MSS received research support from Bayer, Fielmann, Topcon, the German Federal Ministry of Education and Research, the Else Kröner Fresenius Foundation, the Danger Foundation, the Christiansen Foundation and the Claire-Jung Foundation as well as speakers' honoraria/ consulting fees from Abbvie, Alcon, Apellis, Astellas, Atheneum, Bayer, Heexal, Nordic Pharma, Novartis, Roche, SHS, Stada, and TelemedC. CG has nothing to disclose. OMD is supported by the Arizona Department of Health Services (ADHS14-052688). VB is a consultant for GenSight Biologics and Topcon Medical and is supported in part by the National Institutes of Health's National Eye Institute core grant P30-EY06360 (Department of Ophthalmology, Emory University School of Medicine) and by a departmental grant from Research to Prevent Blindness (New York, NY). BG has nothing to disclose. AHA has received honoraria for advice

## Evidence review plan

This study is prospectively registered through PROSPERO (#1154900). We will include randomized controlled trials (RCTs) that enroll patients with non-arteritic CRAO presenting within 4.5 hours of time LKW. We will not include non-controlled interventional studies or retrospective studies. We will search MEDLINE, Embase, the Cochrane Library, Web of Science, and the ClinicalTrials.gov registry from inception through the date of commencement of the systematic review. We will assess the risk of bias using the Cochrane Risk of Bias Tool 2.0. We will contact the corresponding author(s) of any studies identified that meet the study selection criteria. We will inspect, harmonize, and collate trial datasets. The primary end point will be attainment of a final best corrected visual acuity (BCVA) equal to or better than 20/63 (logarithm of the minimum angle of resolution [logMAR] of ≤0.5). Secondary end points will include shift analyses of key visual acuity outcome categories according to the World Health Organization (WHO) International Classification of Disease (ICD)-11, final BCVA considered as a continuous variable, final BCVA equal to or better than 20/100 (logMAR of ≤0.7), final BCVA equal to or better than 20/200 (logMAR of ≤1), a quantitative measure of visual field function (where available), global disability (modified Rankin Scale score [mRS]), and key safety end points (including symptomatic intracranial hemorrhage [sICH] and other systemic hemorrhage). We will fit a series of mixed logistic and linear regression models with trial, and trial by treatment interaction terms as random effects. To probe for sources of heterogeneity, we will pursue a series of subgroup and sensitivity analyses. Finally, a GRADE assessment will be presented.

## Conclusions and relevance

Completion of the proposed study will permit a synopsis of the interventional literature on IV thrombolysis for CRAO, generate a pooled estimate of the treatment effect, and allow exploration of sources of heterogeneity. Such results may be of interest to healthcare professionals, guideline development bodies, policymakers, payors, and future patients with CRAO.

## Registration

This systematic review is prospectively registered: PROSPERO #1154900.

## Introduction

### Background

Central retinal artery occlusion (CRAO) is a devastating form of acute ischemic stroke [1] that usually causes permanent visual loss in one eye. While a majority of acute ischemic strokes affect the brain (cerebral ischemic stroke), a small minority (<1%) affect the retina and – although disabling – are typically recognized later than

or lecturing from BMS/Pfizer, Abbvie, Teva, Novartis, Lilly, Lundbeck and Teva and research grants from the Norwegian Program for Clinical Therapy Research in the specialist health service (Klinbeforsk), the South-Eastern Norway Regional Health Authority, EU, The Norwegian Health Association, Odd Fellow, BMS, Pfizer, and Boehringer-Ingelheim. SP received research support from BMS/Pfizer, Boehringer-Ingelheim, Daiichi Sankyo, European Union, German Federal Joint Committee Innovation Fund, and German Federal Ministry of Education and Research, Helena Laboratories and Werfen as well as speakers' honoraria/consulting fees from Alexion, AstraZeneca, Bayer, Boehringer-Ingelheim, BMS/Pfizer, Daiichi Sankyo, Portola, and Werfen. MS is supported by the National Institutes of Health (R56AG074279, K76AG060001, R01AG078803 and R21AG070859). The above mentioned commercial affiliations do not alter our adherence to PLOS ONE policies on sharing data and methods.

cerebral ischemic strokes. When the retina's main source of metabolic substrates – the central retinal artery – is occluded, rapid loss of visual function ensues. Only a minority of affected patients recover a best-corrected visual acuity (BCVA) of 20/100 or better [2], resulting in long-term quality of life impairments with respect to overall social functioning, employment, mental health, and level of dependency, and a propensity to falls in the elderly [3,4].

### Clinical framework

Intravenous (IV) tissue plasminogen activator (tPA; which almost exclusively refers to alteplase [5] or tenecteplase [6]) is effective in improving functional status after cerebral ischemic stroke [7]. When a clot blocks an artery, IV tPA can break up the clot, restore blood flow (i.e., achieve reperfusion) and increase the chance of salvaging blood-starved tissue [8]. The odds of reperfusion are associated with a the size of the occluded blood vessel and the length of the occluding particle [9]. The central retinal artery has a diameter of only 160um at the optic nerve head [10,11] and is thus ideally poised to benefit from IV tPA. The risk of developing atrial fibrillation (AF; the most common arrythmia in the US and a cause of blood clots) after CRAO is equivalent to that seen in matched patients with cerebral ischemic stroke and higher than in controls [12]. Furthermore, patients with CRAO also have simultaneous cerebral ischemic stroke approximately 20–30% of the time when magnetic resonance imaging (MRI) of the brain is performed concurrently) [13–15], although overt neurological symptoms occurring in tandem with CRAO are less common. These observations argue that similar processes may underpin both disorders.

A recent systematic review and meta-analysis using a standardized definition of visual recovery (attainment of a final BCVA of ≥20/100) found that use of IV tPA improves the proportion of acute CRAO patients with a favorable outcome [16]. In this study, 37.3% of patients treated with IV tPA recovered functional vision in the affected eye, compared with 17.7% of untreated patients. The risk of symptomatic intracranial hemorrhage (the most feared side-effect of IV tPA) was low, with no published cases when treatment was given within 4.5 hours of time last known well (LKW) at the time of this analysis. In the TenCRAOS clinical trial, there was one sICH in a participant that was treated with tenecteplase (presented at ESOC in May 2025).

### Biological framework

The retina requires a constant supply of substrates to maintain tissue viability [17]. As the chief conduit for these metabolic substrates, if the central retinal artery is blocked retinal viability is temporarily maintained by i) passive diffusion of oxygen from the outer retina and choroid and ii) collateral flow from the choroidal circulation. By analogy to cerebral ischemic stroke, there is a retinal "ischemic penumbra" with observed anoxic, hypoxic and normoxic compartments [18]. Pre-clinical data support 4.5 hours as a rational time cutoff for IV tPA administration in CRAO. In non-human primates, occlusion of the central retinal artery proximal to its entry into optic nerve for more than 105 minutes causes severe retinal tissue injury [19–21]. However, in older animals with experimentally-induced chronic hypertension and an atherogenic

diet, irreversible tissue loss does not occur until 240 minutes after occlusion [22]. This suggests that in elderly people at high vascular risk, the retina may be more robust to transient impairments in blood flow [23].

The potential benefit of IV thrombolysis for CRAO remains an area of active discussion. Concerns about the use of IV thrombolysis in CRAO include the possibility that IV thrombolysis does not reach the occluded retinal vessel in meaningful concentrations, that collateralization may be less robust that in the cerebral circulation, and that other mechanisms of CRAO – including calcific emboli, intimal dissection, or arteritic CRAO – would not benefit from IV thrombolysis [24].

### Rationale for individual participant data meta-analysis (IPMA)

Three randomized, controlled, trials (RCTs) enrolling patients with acute non-arteritic CRAO are either complete or at an advanced stage of conduct: THEIA, TenCRAOS, and REVISION. The THEIA trial (NCT03197194) randomized 70 patients with acute CRAO within 4.5 hours of time LKW to either IV alteplase or 300 mg acetylsalicylic acid (ASA) delivered orally. A higher proportion of patients randomized to IV thrombolysis achieved the primary end point but results were not statistically significant [25]. The TenCRAOS trial (NCT04526951) is at an advanced stage of conduct and preliminary results were presented at the European Stroke Organization Conference (ESOC) in May 2025. The REVISION trial (NCT04965038) is actively enrolling with a potential sample size up to 422 participants, and approaching the first interim analysis (120 participants) as of the time of this submission. In the early phase of REVISION, alteplase was the thrombolytic employed, the protocol was amended to use tenecteplase instead. The primary comparator in REVISION is IV placebo.

An IPMA confers several benefits that are relevant to the present clinical issue. First, it would permit derivation of a pooled estimate of treatment effect from the 3 trials. Such an estimate is expected to be more precise than an estimate derived from any individual RCT. Secondly, each trial may individually be considered underpowered to detect a small but clinically meaningful effect size. Pooling data from the three trials will enhance statistical power to assess the effect of IV tPA on clinical outcomes. Third, an IPMA will allow exploration of individual patient-level sources of heterogeneity in a manner that would not be possible in a study-level meta-analysis (owing to the phenomenon of ecological fallacy). Fourth, signals of safety concerns are more likely to manifest in a larger cohort of study participants [26] and may provide important context to pooled estimates of efficacy. Finally, IPMA is considered the highest echelon of evidence with respect to evidence grading and this study may be of use to professional bodies in updating guideline recommendations for CRAO [27].

### Study objectives

Constituent study objectives are outlined in Table 1 (overleaf). The primary study objective is to address the following question, reflecting a common dilemma in clinical practice:

*"In adults ≥18 years old with acute, non-arteritic, CRAO and disabling visual symptoms (a BCVA of worse than 20/100 at onset; underline population), does treatment with IV thrombolysis (either alteplase or tenecteplase) within 4.5 hours of time LKW (underline intervention) increase the odds of achieving a final BCVA of ≥20/63 (underline outcome) when compared with no IV thrombolysis, placebo, and/or oral anti-thrombotic therapy (underline comparator)?"*

## Methods

### Study design

This systematic review and IPMA is prospectively registered through PROSPERO (#1154900). We will report it in accordance with the PRISMA-IPD reporting guideline [28,29]. Case record forms from individual trials, analytic code, and

**Table 1. Study Objectives.**

| Systematic Review |
| --- |
| ● Primary Objective: |
| 1. To identify all randomized, controlled trials within the world literature that meet selection criteria to provide the most precise pooled estimate of treatment effect. |
| ● Secondary Objective: |
| 1. To critically appraise the methodology of included studies according to standardized criteria, assess risk of bias and contextualize results of a meta-analysis. |
| **Individual Participant Data Meta-Analysis** |
| ● Primary Objective: |
| 1. To derive a pooled estimate of the effect of IV thrombolysis on visual outcomes in patients with acute non-arteritic CRAO. |
| ● Secondary Objectives: |
| 1. To explore study- and participant-level source of heterogeneity. |
| 2. To determine whether IV thrombolysis is related to key safety outcomes. |
| 3. To determine whether there is a relationship between time to treatment with IV thrombolysis and visual outcomes. |

**Acronyms:** CRAO, central retinal artery occlusion; IV, intravenous.

summary results of individual clinical trials will be disseminated as **Supplemental Material** to the main manuscript. The extracted data from studies may be made available upon request to the corresponding authors of each individual clinical trial, in accordance with any applicable regulations and data sharing policies. No automation tools will be used in the screening or data extraction processes. This study has been deemed exempt by the Duke Institutional Review Board (Pro00117992) as this is deemed non-human subjects research given the de-identified nature of the underlying data.

## Eligibility criteria

We will include only RCTs that enroll patients with acute, non-arteritic CRAO who are randomized and treated within 4.5 hours of time LKW. We will permit inclusion of trials where the comparator is no IV thrombolysis, placebo, and/or oral anti-thrombotic therapy. RCTs that also permit the deployment of conservative therapies [30] will only be included if use of such therapies is permitted in both arms (treatment and control). We will exclude trials that enrolled fewer than 20 patients (because such small trials tend to entail bias and provide unstable estimates of treatment effect) as well as any trials where IV thrombolysis was given alongside another trial-mandated intervention (medical or surgical therapy).

## Information sources and search strategy

We will search MEDLINE (via Ovid), Embase (via Elsevier), the Cochrane Library (via Wiley), Web of Science – Science Citation Index and Social Science Citation Index (via Wiley), and the ClinicalTrials.gov registry from inception of each registry through June 30th, 2026. We will use database-specific subject headings and keywords related to IV thrombolysis, IV tPA, alteplase, tenecteplase, and CRAO (and its permutations). No restrictions will be placed on the publication year. We will place no limitations on language. An experienced medical librarian will devise and conduct the searches, with input on keywords from other authors. The search strategy will be independently peer-reviewed by another librarian using a modified Peer Review of Electronic Search Strategies (PRESS) checklist [31]. We will hand-search the bibliographies of included studies. We will not search literature produced outside of traditional publishing avenues such as unpublished conference materials, theses, government or commercial white papers, or technical reports (sometimes referred to as the "grey literature"). Completed trials identified through ClinicalTrials.gov will be included). The full, reproducible search strategy for each included database will be disseminated as part of the **Supplemental Material** to our primary manuscript.

## Screening process

All unique citations will be imported into Covidence [32], a web-based systematic review platform for screening. Two investigators will independently screen all titles and abstracts. Next, potentially eligible records will be independently screened by two investigators. At both stages, conflicts will be resolved through discussion and arbitration by a third study member where necessary.

## Data collection process

Once all trials that meet our study criteria re identified, a minimum dataset will be designed by the collaborating authors and reviewed by the present study's statistician. Subsequently, this minimum dataset will be provided to each trial statistician. Trial statisticians will extract participant-level data by direct access to study databases. At a minimum, fully de-identified data from the THEIA, TenCRAOS, and REVISION trials will be provided to Duke University under a data transfer and utilization agreement (DTUA) between Duke University and the primary institution associated with each RCT. If further RCTs that meet our selection criteria are identified, we will contact the corresponding author of the trial manuscript to request individual participant-level data.

Two investigators will independently inspect, clean, and collate the datasets. Indicator variables will be created where necessary. All BCVA measurements will be harmonized and converted to logMAR per Table 2 (overleaf). Discrepancies identified at this point will be resolved through discussion, and we will maintain communication with each trial's primary statistician in case queries arise during data cleaning and collation. A single data frame will be created from the pooled source trial datasets containing indicator variables identifying the source trial for each participant. The resulting data frame will represent the master dataset for the present study, and imputation of missing data will only be performed after this step. As a form of quality control, descriptive and inferential statistics from the parent RCT will be re-calculated and cross-checked against conference presentation results and any corresponding publications.

## Data items

At a minimum, the following data elements will be included in our study:

1) **Study-Level Parameters**

Study design, blinding method, comparator, randomization procedure, country/countries of study conduct, total number of patients included.

2) **Participant-level parameters**

● Baseline Attributes:

Age, biological sex, National Institutes of Health Stroke Scale (NIHSS) score on presentation, presenting BCVA, past medical history (including hypertension, hyperlipidemia, diabetes mellitus, atrial fibrillation, tobacco exposure, and chronic kidney disease). We will collect data on race and ethnicity, where possible, for descriptive purposes. However, we anticipate that because of legal restriction in certain countries where constituent trials are conducted, this will likely not be possible for all participants. Therefore, we do not anticipate being able to include these parameters within any modeling or subgroup analyses.

● Treatment Parameters:

Time of symptom onset, time to emergency department (ED) presentation, time of randomization, treatment assignment (intervention versus control), time of intervention delivery (if delivered successfully), IV thrombolytic agent used (alteplase

**Table 2. Best-corrected visual acuity conversion algorithm (Adapted from Schulze-Bonsel et al., 2006 [33]).**

| Decimal Acuity | logVA | logMAR | FrACT [34] |
|---|---|---|---|
| 2.00 | 0.30 | −0.30 | |
| 1.58 | 0.20 | −0.20 | |
| 1.26 | 0.10 | −0.10 | |
| 1.00 | 0.00 | 0.00 | |
| 0.79 | −0.10 | 0.10 | |
| 0.63 | −0.20 | 0.20 | |
| 0.50 | −0.30 | 0.30 | |
| 0.40 | −0.40 | 0.40 | |
| 0.32 | −0.50 | 0.50 | |
| 0.25 | −0.60 | 0.60 | |
| 0.20 | −0.70 | 0.70 | |
| 0.16 | −0.80 | 0.80 | |
| 0.13 | −0.90 | 0.90 | |
| 0.10 | −1.00 | 1.00 | |
| 0.079 | −1.10 | 1.10 | |
| 0.063 | −1.20 | 1.20 | |
| 0.050 | −1.30 | 1.30 | |
| 0.040 | −1.40 | 1.40 | |
| 0.032 | −1.50 | 1.50 | |
| 0.025 | −1.60 | 1.60 | |
| 0.020 | −1.70 | 1.70 | |
| 0.016 | −1.80 | 1.80 | |
| 0.013 | −1.90 | 1.90 | CF* |
| 0.010 | −2.00 | 2.00 | |
| 0.0079 | −2.10 | 2.10 | |
| 0.0063 | −2.20 | 2.20 | |
| 0.0050 | −2.30 | 2.30 | HM* |
| 0.0040 | −2.40 | 2.40 | |
| 0.0032 | −2.50 | 2.50 | |
| 0.0025 | −2.60 | 2.60 | |
| 0.0020 | −2.70 | 2.70 | LP* |
| 0.0016 | −2.80 | 2.80 | |
| 0.0013 | −2.90 | 2.90 | |
| 0.0010 | −3.00 | 3.00 | NLP* |

**Acronyms:** CF, count fingers, FrACT, Freiburg visual acuity test; HM, hand motion; logMAR, logarithm of the minimum angle of resolution; logVA, logarithm of the visual acuity; LP, light perception; NLP, no light perception.

*The conversion of these measurements to numeric values is somewhat controversial [35]. However, this will be only relevant for our secondary analysis in which final BCVA is considered as a continuous variable. For the remainder of the analyses, these will be below the BCVA threshold defining our primary end point.

or tenecteplase), dose of IV thrombolytic agent received, concurrent therapies employed (including ocular massage, hemodilution, anterior chamber paracentesis, topical or systemic intraocular pressure-lowering drugs, or hyperbaric oxygen therapy), and control/active control used (placebo, anti-thrombotic therapy, or other).

● End Points:

Follow-up BCVA, final BCVA, modified Rankin Scale score (mRS), symptomatic intracranial hemorrhage (sICH; per the modified Safe Implementation of Thrombolysis in Stroke-Monitoring Study [SITS-MOST] definition which includes any new intracranial hemorrhage associated with a decline of ≥4 points on the National Institutes of Health Stroke Scale (NIHSS) score [36]) occurring within 36 hours of thrombolysis administration, any ICH, orolingual angioedema, gastrointestinal hemorrhage, other extracranial hemorrhage, and mortality. We will ascertain the precise definition of each complication employed as we anticipate that this will vary by trial.

## Risk of bias in individual studies

Two investigators will independently assessed risk of bias in individual studies using the Cochrane Risk of Bias 2.0 tool for interventional studies [37]. Discrepancies in risk of bias assessments will be resolved through discussion. Risk of bias will be presented in graphical and narrative form using established methodology.

## Synthesis methods

Key participant, and study-level attributes ("Data Items", above) will be disaggregated by intervention status (treatment versus control), presented in tabular form, and synthesized narratively. Counts and percentages will be used to summarize categorical variables and medians with first (Q1) and third (Q3) quartiles will be used to summarize the distribution of continuous variables. We will calculate absolute standardized differences to illustrate differences across groups with respect to baseline participant- and study-level attributes.

**Primary end point.** The primary end point will be attainment of a final BCVA equal to or better than 20/63 (logMAR of ≤0.5). This threshold was chosen because i) it indicates a patient-centered degree of visual recovery (classed at the lower threshold of "mild visual impairment" by the WHO), ii) because the requirement for this degree of recovery introduces a conservative bias in to the analysis, and iii) because an apparent improvement to 20/63 cannot be achieved via eccentric fixation. We will conduct a one-stage IPMA to derive a pooled estimate of the treatment effect of IV tPA. For this primary analysis, a mixed effects binary logistic regression model will be fit. The dependent variable will be a final BCVA of ≥20/63 [logMAR 0.50]. The independent variable will be treatment assignment (IV thrombolysis or control, intention-to-treat basis). We will include presenting BCVA and stage of trial (first half or second half) as fixed effects. We will include two random effects: i) an indicator for trial and ii) a trial-by-treatment interaction term. Unadjusted and adjusted odds ratios with their 95% confidence intervals (CI) will be computed. The p-value corresponding to the adjusted model will be computed (and will be the only p-value presented in the manuscript outside of interaction analyses). Risk differences will be calculated as differences of predicted proportions from binary logistic regression models. Should the assumption of logistic regression not hold, we will implement transformations as appropriate or use an alternative regression strategy.

**Secondary end points.** Secondary end points will include final BCVA (considered as a continuous variable), a shift analysis of key VA outcome categories according to the WHO ICD-11 ("normal vision" (logMAR ≤ 0.0), "mild vision impairment" (logMAR > 0.0 and ≤ 0.5), "moderate vision impairment" (logMAR > 0.5 and ≤ 1.0), "severe vision impairment" (logMAR > 1.0 and ≤ 1.3), "counting fingers" (logMAR > 1.3 and ≤ counting fingers), "hand motion or light perception", and "no light perception"), final BCVA ≥20/100 (logMAR ≤ 0.7), final BCVA ≥20/200 (logMAR of ≤1.0), a quantitative measure of visual field function (where available), global disability (mRS), and key safety end points (including sICH, other systemic hemorrhage, and orolingual angioedema).

For final BCVA treated as a continuous variable, a mixed effects generalized linear model will be fit. The dependent variable will be final BCVA. The independent variable will be treatment assignment (IV thrombolysis or control, intention to-treat basis). We will include presenting BCVA, and stage of trial (first half or second half) as a fixed effects and trial and a trial-by-treatment interaction term as random effects. A beta coefficient, 95% confidence interval (CI), and corresponding

 

p-value will be computed. Should the assumption of linear regression not hold, we will implement transformations as appropriate or use an alternative regression strategy.

Other secondary end points will be modelled using similar statistical methodology. For binary outcomes (e.g., final BCVA of ≥20/100 [logMAR ≤ 0.70], three-line on-chart improvement in BCVA, sICH), mixed effects binary logistic regression modelling will be used. For ordinal outcomes (e.g., ordinal categories of BCVA outcomes and mRS), mixed effects ordinal logistic regression modelling will be used. We anticipate that safety end points other than sICH (including orolingual angioedema) will be reported using varying definitions and be of low incidence. Therefore it will likely not be possible to fit appropriate models. We plan to synthesize these end points narratively and in tabular form.

**Subgroup analyses.** To explore possible sources of heterogeneity, we will perform a limited number of pre-specified subgroup analyses based on biologically plausible individual participant parameters with adjustment for multiplicity (using Bonferroni correction). To explore possible participant-level causes of heterogeneity, we will perform subgroup analyses according to 1) the form of IV thrombolysis used (alteplase versus tenecteplase), 2) age (<65 or ≥65 years of age), 3) biological sex, 4) presenting BCVA (logMAR 0.80 to 1.70 versus "Count Fingers or Worse" [logMAR ≥ 1.80]), and 5) treatment windows to include 0–1.5 hours, 1.5–3 hours, and 3–4.5 hours. If any identified trials employed stratification variables, we will include them as subgroups within the pooled cohort. If a sufficient number of trials are identified, study-level subgroups will be defined according to: 1) study risk of bias and 2) blinding method.

Finally, to attempt to partially overcome some of the pitfalls of individual covariate-defined subgroup analysis, we will use multiple logistic regression to derive model-defined subgroups and identify potentially therapeutically-relevant categories for future study [38–41].

**Supplemental/Sensitivity analyses.** We will pursue the following supplemental/sensitivity analyses:

1) We will replicate our primary analysis on a "per-protocol" framework as opposed to an "intention-to-treat" framework.

2) We will replicate our primary analysis only including those participants with at least a 6-month visual recovery measurement recorded.

3) We will replicate the primary analysis without adjustment for presenting BCVA.

4) We will create a multiple logistic regression model (with covariates that are plausibly expected to be independently associated with BCVA [including key demographic, clinical, and treatment-related factors] as fixed effects and trial as a random effect). Then, we will derive scores reflecting the propensity to be assigned to either intervention or control. We will repeat the primary analysis using the technique of propensity score overlap weighting [42–44], a technique in which participants are weighted in proportion to the probability that they would belong to the opposite group given their measured attributes.

5) To assess for sensitivity to model structure, we will repeat the above analysis using conventional multiple regression with the same covariates.

6) We will conduct an influence analysis in which we leave one study out at a time and replicate our primary analysis.

7) We will conduct a two-stage IPMA with the overall objective to derive a similar pooled estimate (including an influence analysis as per point #5 above).

8) Further sensitivity analyses pertaining to the treatment of missing data are detailed under the "Missing Data" section (below).

**Time-to treatment and final BCVA.** Final BCVA as a function of time-to-randomization will be modelled using a mixed effects generalized linear model. This model will include first recorded BCVA as a fixed effect and trial and a

trial-by-treatment interaction term as random effects. Because this primary model will include both treatment groups (treated and untreated), an additional time-by-treatment interaction term will be included. If the relationship is non-linear, time-to-randomization will be modelled as a restricted cubic spline.

To further explore the interplay of time-to-treatment and BCVA, nonlinear models will be fit exploring the relationship between time and final BCVA divided in to four categories based on the World Health Organization (WHO)-defined categories: 0.3–0.5 logMAR (mild vision impairment), 0.5–1.0 logMAR (moderate vision impairment), 1.0–1.3 logMAR (severe vision impairment), and 1.3–2.9 logMAR (blindness) [45]. The relationship between time and these categories will be examined in three separate mixed effects binary logistic regression models using a locally weighted scatterplot smoothing (LOWESS) technique. A common odds ratio of the ordinal shift in the distribution of visual function over the categories presented will be estimated from these models. By averaging values derived from the algorithmic joint outcome table, we will determine the proportion of patients having a better outcome by one category using methodology previously applied to other ordinal scales within stroke medicine [46]. For binary outcomes, the probability of each will be modelled as a function of time using a mixed effects binary logistic regression model.

**Missing data.** Missing data will be assumed to be missing completely at random. Our primary missing data approach will be to use multiple imputation via chained equations (MICE). As supplemental analyses, we will perform a complete case analysis, and last observation carried forward (for our primary end point). We will also perform "best-case" and "worst-case" imputation on the primary end point and present the point estimates/95% CIs derived from the new dataset.

**General considerations.** In general, to present treatment effects across groups, point estimates and their 95% CIs will be presented. A two-sided alpha of 0.05 will be considered statistically significant. No adjustment for multiple comparisons will be made. All analyses will be performed using the *rms*, *hmisc*, *meta*, *metafor*, and *ggplot2* packages in R (R Foundation for Statistical Computing, Vienna, Austria, v. 4.1.2).

## Publication bias assessment

To investigate the presence of reporting biases/small study effects, we will create funnel plots for each outcome and visually inspect the plot for evidence of asymmetry. This is likely to be highly exploratory given the low numbers of anticipated studies in this analysis.

## Grades of recommendation assessment development and evaluation (GRADE) assessment

We will prepare an evidence profile, assess the overall quality of the evidence, and offer a recommendation for future clinical practice using the GRADE framework [47].

## Dissemination plan

We will submit a manuscript arising from this analysis to an international, peer-reviewed journal and present the results at one or more scientific conferences (such as the International Stroke Conference, European Stroke Organization Conference, or World Stroke Congress).

## Discussion

Acute CRAO is a disabling subtype of acute ischemic stroke that results in severe, typically permanent, visual loss. At present, there are no evidence-based therapies for this condition. A series of retrospective, observational, cohort studies (and meta-analyses of these studies) have suggested a positive association between early IV thrombolysis and visual recovery. While in general well-designed observational studies do not produce larger treatment effect estimates than those observed in RCTs [48–50], the literature to-date is of low-to-moderate quality with a high likelihood of selection bias, ascertainment bias, and confounding. The three RCTs (THEIA, TenCRAOS, and REVISION), which are either completed

or at an advanced stage of conduct, will help fill the evidence gap with high-quality data on IV thrombolysis and visual outcomes in acute CRAO.

The planned IPMA will include several advantages. First, a pooled estimate will likely confer greater precision than that obtained from any individual RCT. Second, the use of IPMA will permit the exploration of participant-level sources of heterogeneity of treatment effect. Third, an individual participant-level approach will permit flexibility in our modelling approach and sensitivity analyses that appropriately examine whether our approach is robust to key assumptions. Finally, a structured appraisal of the literature using a GRADE approach will provide data on the scope of the evidence, the overall likelihood of bias, the magnitude of any effect observed, and the potential relevance of these data to guideline generation.

The present study will also allow a comparison of the treatment effect of tenecteplase and alteplase in patients with acute CRAO. Tenecteplase is a newer-generation thrombolytic agent that has favorable pharmacological properties [51–55] and is non-inferior to alteplase with respect to safety and long-term functional outcomes when used in patient with acute cerebral ischemic stroke [56–59]. Because it can be delivered as a single bolus, it is ideally suited to the treatment of patients with acute CRAO, whose complex needs demand a highly efficient, coordinated approach to emergency treatment. THEIA used exclusively alteplase, TenCRAOS used exclusively tenecteplase, while REVISION allocated participants to alteplase (versus placebo) in the earliest phase but converted to tenecteplase (versus placebo) in response to widespread practice pattern changes in favor of tenecteplase. Thus, we anticipate that there will be a sufficient number of treated patients to permit a reasonable statistical analysis (although, as with all subgroups analyses, power may remain a potential pitfall).

There are several pitfalls to the potential analysis. First, although IPMA represents the highest echelon of evidence, issues of unmeasured differences between treatment groups, differences in study methodology, and heterogeneity of underlying populations from which participants are drawn remain. Second, it will likely not be possible to include race and ethnicity as covariates within our modelling, a limitation because there are known associations between these critical parameters and ischemic stroke outcomes [60]. This is because the three trials are conducted in Europe, where a low proportion of patients identify as Black, and because ascertaining data on race or ethnicity is not legally permitted in certain jurisdictions where the trials are conducted. Third, while a secondary objective is to determine whether there is a signal of harm (especially an increased risk of sICH) associated with IV thrombolysis in this population, analyses of safety end points are likely to remain underpowered, a general property of this form of analysis. However, it is estimated that the risk of sICH is low in patients with acute CRAO, likely approximating that seen in minor cerebral ischemic stroke (in the case of concurrent, silent, cerebral ischemia) or in "stroke mimics" [61,62] when there is not concurrent cerebral ischemia. Fourth, we have chosen to focus on the 4.5 hour window from time last known well to treatment. The rationale for this choice is i) it is our anticipation that all trials examined as part of our systematic review will adopt this threshold and ii) these results will be of the most immediate translational relevance. However, the results obtained cannot be generalized to those patients who, in future, might be considered for treatment beyond 4.5 hours of time last known well based on estimates of potential efficacy using markers of retinal structure and function that could be used to infer retinal viability in select patients. Finally, subgroup analyses must be approached with caution, given the risks of type I error and insufficient power to detect clinically relevant between-group differences. To address this, we plan only a small number of pre-specified, biologically plausible subgroup analyses with correction for multiplicity. Furthermore, we supplement this with a modern, multiple regression modelling-based approach to make the highest quality inferences possible in subgroup data.

## Conclusions

Completion of the proposed study will permit a synopsis of the interventional literature on IV thrombolysis for CRAO, generate a pooled estimate of the treatment effect, and allow exploration of sources of heterogeneity. Such results may be of interest to healthcare professionals, guideline development bodies, policymakers, payors, and future patients with CRAO.

## Author contributions

**Conceptualization:** Brian Mac Grory, Cécile Preterre, Pierre Lebranchu, Stephen James Ryan, Øystein Kalsnes Jørstad, Morten Carstens Moe, Johannes Tünnerhoff, Martin S. Spitzer, Carsten Grohmann, Oana M. Dumitrascu, Valérie Biousse, Benoit Guillon, Anne Hege Aamodt, Sven Poli, Matthew Schrag.

**Data curation:** Brian Mac Grory, Cécile Preterre, Pierre Lebranchu, Stephen James Ryan, Øystein Kalsnes Jørstad, Morten Carstens Moe, Johannes Tünnerhoff, Martin S. Spitzer, Carsten Grohmann, Oana M. Dumitrascu, Valérie Biousse, Benoit Guillon, Anne Hege Aamodt, Sven Poli, Matthew Schrag.

**Investigation:** Brian Mac Grory.

**Methodology:** Brian Mac Grory.

**Project administration:** Brian Mac Grory.

**Resources:** Brian Mac Grory.

**Supervision:** Valérie Biousse, Anne Hege Aamodt, Sven Poli, Matthew Schrag.

**Writing – original draft:** Brian Mac Grory.

**Writing – review & editing:** Brian Mac Grory, Cécile Preterre, Pierre Lebranchu, Stephen James Ryan, Øystein Kalsnes Jørstad, Morten Carstens Moe, Johannes Tünnerhoff, Martin S. Spitzer, Carsten Grohmann, Oana M. Dumitrascu, Valérie Biousse, Benoit Guillon, Anne Hege Aamodt, Sven Poli, Matthew Schrag.

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
