## [Decision Letter · Decision Letter 0]

22 Dec 2025

Dear Dr. Mac Grory,

Thank you for submitting your manuscript to PLOS ONE. After careful consideration, we feel that it has merit but does not fully meet PLOS ONE’s publication criteria as it currently stands. Therefore, we invite you to submit a revised version of the manuscript that addresses the points raised during the review process.

We look forward to receiving your revised manuscript.

Kind regards,

Ogugua Ndubuisi Okonkwo, M.D.

Academic Editor

PLOS One

Journal Requirements:

2. In your cover letter, please confirm that the research you have described in your manuscript, including participant recruitment, data collection, modification, or processing, has not started and will not start until after your paper has been accepted to the journal (assuming data need to be collected or participants recruited specifically for your study). In order to proceed with your submission, you must provide confirmation.

“BMG is supported by the National Institutes of Health (K23HL161426, R03HL178686, and UG3NS138219), the American Heart Association (23MRFSCD1077188 & 25GLP1450119), Duke Bass Connections, the Duke Office of Physician-Scientist Development, and the Duke University Office of the Provost.

CP has nothing to disclose.

PL received speakers’ honoraria/consulting fees from Santhera, Novartis, Lissac-Optic2000, Alexion, and Amgen.

SJR has nothing to disclose.

ØKJ: Allergan (speaker, consultant), Bayer (speaker, consultant), Chiesi Farmaceutici (speaker), Roche (consultant), and SJJ Solutions (speaker, consultant, and royalties).

MCM Allergan (consultant), Bayer (speaker, consultant), Roche (speaker, consultant), Apellis (consultant), Novartis (consultant) and SJJ Solutions (consultant, and royalties).

JT has nothing to disclose.

MSS received research support from Bayer, Fielmann, Topcon, the German Federal Ministry of Education and Research, the Else Kröner Fresenius Foundation, the Danger Foundation, the Christiansen Foundation and the Claire-Jung Foundation as well as speakers’ honoraria/consulting fees from Abbvie, Alcon, Apellis, Astellas, Atheneum, Bayer, Heexal, Nordic Pharma, Novartis, Roche, SHS, Stada, and TelemedC.

CG has nothing to disclose.

OMD is supported by the Arizona Department of Health Services (ADHS14-052688).

VB is a consultant for GenSight Biologics and Topcon Medical and is supported in part by the National Institutes of Health’s National Eye Institute core grant P30-EY06360 (Department of Ophthalmology, Emory University School of Medicine) and by a departmental grant from Research to Prevent Blindness (New York, NY).

BG has nothing to disclose.

AHA has received honoraria for advice or lecturing from BMS/Pfizer, Abbvie, Teva, Novartis, Lilly, Lundbeck and Teva and research grants from the Norwegian Program for Clinical Therapy Research in the specialist health service (Klinbeforsk), the South-Eastern Norway Regional Health Authority, EU, The Norwegian Health Association, Odd Fellow, BMS, Pfizer, and Boehringer-Ingelheim.

SP received research support from BMS/Pfizer, Boehringer-Ingelheim, Daiichi Sankyo, European Union, German Federal Joint Committee Innovation Fund, and German Federal Ministry of Education and Research, Helena Laboratories and Werfen as well as speakers’ honoraria/consulting fees from Alexion, AstraZeneca, Bayer, Boehringer-Ingelheim, BMS/Pfizer, Daiichi Sankyo, Portola, and Werfen.

MS is supported by the National Institutes of Health (R56AG074279, K76AG060001, R01AG078803 and R21AG070859).”

We note that one or more of the authors are employed by a commercial company

4. In the online submission form you indicate that your data is not available for proprietary reasons and have provided a contact point for accessing this data. Please note that your current contact point is a co-author on this manuscript. According to our Data Policy, the contact point must not be an author on the manuscript and must be an institutional contact, ideally not an individual. Please revise your data statement to a non-author institutional point of contact, such as a data access or ethics committee, and send this to us via return email. Please also include contact information for the third party organization, and please include the full citation of where the data can be found.

Reviewers' comments:

Reviewer's Responses to Questions

**Comments to the Author**

1. Does the manuscript provide a valid rationale for the proposed study, with clearly identified and justified research questions?

Reviewer #1: Yes

Reviewer #2: Yes

2. Is the protocol technically sound and planned in a manner that will lead to a meaningful outcome and allow testing the stated hypotheses?

Reviewer #1: Yes

Reviewer #2: Yes

3. Is the methodology feasible and described in sufficient detail to allow the work to be replicable?

Reviewer #1: Yes

Reviewer #2: Yes

4. Have the authors described where all data underlying the findings will be made available when the study is complete?

Reviewer #1: Yes

Reviewer #2: Yes

5. Is the manuscript presented in an intelligible fashion and written in standard English?

*PLOS ONE*

Reviewer #1: Yes

Reviewer #2: Yes

You may also provide optional suggestions and comments to authors that they might find helpful in planning their study.

Reviewer #1: Authors present the protocol for a systematic review and individual patient data meta-analysis of trials which randomized patients with CRAO to intravenous thrombolysis. There is currently no approved acute revascularization treatment for CRAO and off-label treatments in everyday clinical practice are widely used with great uncertainty. Although there is a need for robust high-quality evidence for supporting such treatment decisions, patient recruitment in trials of acute treatments in CRAO is significantly limited because its occurrence is not as frequent as acute ischemic stroke and because of hospital admission delays. It is unlikely that any single RCT will be powered to demonstrate statistically significant but relevant visual functional improvement. Here lies the importance of this IPDMA, which will probably influence treatment guidelines worldwide. I congratulate the authors for this cooperation effort.

I have only minor comments:

1. Studies performing MRI in patients with CRAO show a relatively high proportion of patients with silent acute ischemic lesions, but concurrent ischemic stroke (i.e. clinical neurological symptoms associated with acute ischemic lesions) is rare. This should be corrected in the introduction.

2. After the recent publication of THEIA, the phrase “with no published cases when treatment was given within 4.5 hours of time last known well” is no longer true. Please correct.

3. Under the paragraph “Rational for individual participant data meta-analysis” please update the part referring to THEIA, which is now published.

4. Please use another term (or sentence) for describing “grey literature”, for readability.

5. Please specific in which time window will sICH be defined (also according to SITS-MOST?).

6. The different trials which will be included in the IPMA appear to have defined the primary endpoint differently. I think it would be useful for the readers that the authors explain why they chose logMAR 0.50 as the primary endpoint, and what it means in terms of visual function (it it described under the section “Secondary Endpoints” but it should be presented under the section “Primary Endpoint”).

7. Please correct “European Stroke Organization annual conference” to European Stroke Organisation Conference”.

Reviewer #2: This registered report protocol is well written, timely, and methodologically robust. The clinical question is highly relevant, the rationale for an IPD meta-analysis is clearly articulated, and the planned analyses are detailed, appropriate, and transparent. I support publication after addressing the following minor points for clarification and completeness:

1. Justification of the 0–4.5 hour time window

You have chosen a ≤4.5 h time frame from last known well (LKW) to IV thrombolysis (IVT), aligned with cerebral ischemic stroke practice and with much of the CRAO thrombolysis literature. However, in CRAO there is an ongoing debate about the role of collateral flow and retinal “penumbra” and whether some patients may retain salvageable tissue beyond 4.5 hours.

• Please expand the rationale for choosing 4.5 hours rather than including trials with treatment up to 24 hours (or other extended windows).

• It would be helpful to explicitly relate this to:

o The non-human primate data suggesting variable retinal tolerance (e.g., the 105 vs 240 minute survival times and how these were interpreted).

o The hypothesis that collateral choroidal flow and individual vascular risk profiles might modulate the true therapeutic window.

o The practical/clinical trade-off between maximizing internal validity (homogeneous early time window) vs external validity (relevance to real-world late presenters).

Even if you ultimately keep the 0–4.5 h window as an inclusion criterion, a more explicit discussion in the Methods and/or Discussion of why late-presenting patients (e.g., up to 24 h) and extended windows are not incorporated would strengthen the protocol and preempt future interpretative debates.

2. More granular onset-to-treatment time subgroup analyses

You already plan to model time-to-treatment and its association with visual outcomes. Given the central importance of timing in CRAO, I would encourage you to pre-specify more granular time windows as part of your subgroup or sensitivity analyses.

For example, you could consider:

• 0–1.5 hours

• 1.5–3 hours

• 3–4.5 hours

• (If data permit, an exploratory >4.5 h subgroup where such patients exist in any contributing trial.)

This does not have to change your primary analysis, which can remain continuous (or modeled with splines), but explicitly listing these time strata as planned exploratory subgroups would:

• Enhance interpretability for clinicians (who often think in discrete time windows).

• Potentially identify clinically meaningful thresholds within the 0–4.5 h window.

• Make it easier to align your findings with how stroke systems of care are organized in practice.

3. Safety endpoints – explicit inclusion of angioedema

In the Methods you already plan to collect several safety endpoints, including symptomatic ICH and systemic hemorrhage. You also mention angioedema later in the text, but it would be helpful to:

• Explicitly list “orolingual angioedema” among the predefined safety endpoints in the main safety endpoint section, alongside sICH and systemic hemorrhage.

• Clarify whether angioedema will be:

o Analysed descriptively only (expected low incidence), or

o Included in any pooled or exploratory safety models, if numbers permit.

Given the known association of IVT with orolingual angioedema, this explicit inclusion will be reassuring both for readers and guideline developers focusing on safety.

**Do you want your identity to be public for this peer review?** For information about this choice, including consent withdrawal, please see our Privacy Policy

Reviewer #1: No

Reviewer #2: **Yes:** Klearchos Psychogios

---

## [Author Response · Author response to Decision Letter 1]

6 Jan 2026

EDITORIAL REQUIREMENTS:

The authors appreciate this comment. The manuscript has been updated to observe these formatting requirements.

2. In your cover letter, please confirm that the research you have described in your manuscript, including participant recruitment, data collection, modification, or processing, has not started and will not start until after your paper has been accepted to the journal (assuming data need to be collected or participants recruited specifically for your study). In order to proceed with your submission, you must provide confirmation.

Thank you, we have revised the cover letter to confirm this.

“BMG is supported by the National Institutes of Health (K23HL161426, R03HL178686, and UG3NS138219), the American Heart Association (23MRFSCD1077188 & 25GLP1450119), Duke Bass Connections, the Duke Office of Physician-Scientist Development, and the Duke University Office of the Provost.

CP has nothing to disclose.

PL received speakers’ honoraria/consulting fees from Santhera, Novartis, Lissac-Optic2000, Alexion, and Amgen.

SJR has nothing to disclose.

ØKJ: AbbVie (speaker, consultant), Bayer (speaker, consultant), Chiesi Farmaceutici (speaker), Roche (consultant), and SJJ Solutions (speaker, consultant, and royalties).

MCM Allergan (consultant), Bayer (speaker, consultant), Roche (speaker, consultant), Apellis (consultant), Novartis (consultant) and SJJ Solutions (consultant, and royalties).

JT has nothing to disclose.

MSS received research support from Bayer, Fielmann, Topcon, the German Federal Ministry of Education and Research, the Else Kröner Fresenius Foundation, the Danger Foundation, the Christiansen Foundation and the Claire-Jung Foundation as well as speakers’ honoraria/consulting fees from Abbvie, Alcon, Apellis, Astellas, Atheneum, Bayer, Heexal, Nordic Pharma, Novartis, Roche, SHS, Stada, and TelemedC.

CG has nothing to disclose.

OMD is supported by the Arizona Department of Health Services (ADHS14-052688).

VB is a consultant for GenSight Biologics and Topcon Medical and is supported in part by the National Institutes of Health’s National Eye Institute core grant P30-EY06360 (Department of Ophthalmology, Emory University School of Medicine) and by a departmental grant from Research to Prevent Blindness (New York, NY).

BG has nothing to disclose.

AHA has received honoraria for advice or lecturing from BMS/Pfizer, Abbvie, Teva, Novartis, Lilly, Lundbeck and Teva and research grants from the Norwegian Program for Clinical Therapy Research in the specialist health service (Klinbeforsk), the South-Eastern Norway Regional Health Authority, EU, The Norwegian Health Association, Odd Fellow, BMS, Pfizer, and Boehringer-Ingelheim.

SP received research support from BMS/Pfizer, Boehringer-Ingelheim, Daiichi Sankyo, European Union, German Federal Joint Committee Innovation Fund, and German Federal Ministry of Education and Research, Helena Laboratories and Werfen as well as speakers’ honoraria/consulting fees from Alexion, AstraZeneca, Bayer, Boehringer-Ingelheim, BMS/Pfizer, Daiichi Sankyo, Portola, and Werfen.

MS is supported by the National Institutes of Health (R56AG074279, K76AG060001, R01AG078803 and R21AG070859).”

We note that one or more of the authors are employed by a commercial company

These requirements have been observed in the revised manuscript and in the cover letter.

4. In the online submission form you indicate that your data is not available for proprietary reasons and have provided a contact point for accessing this data. Please note that your current contact point is a co-author on this manuscript. According to our Data Policy, the contact point must not be an author on the manuscript and must be an institutional contact, ideally not an individual. Please revise your data statement to a non-author institutional point of contact, such as a data access or ethics committee, and send this to us via return email. Please also include contact information for the third party organization, and please include the full citation of where the data can be found.

The aggregated anonymized data will be held by the corresponding author at Duke University, Brian Mac Grory. There is no single external repository of the data as this will be a pooled dataset of randomized controlled clinical trials. The institutional contact is the Duke IRB: https://irb.duhs.duke.edu/.

Thank you, we have observed this requirement.

Thank you, we have observed this requirement.

SCIENTIFIC REVIEWER COMMENTS:

● Reviewer #1:

Authors present the protocol for a systematic review and individual patient data meta-analysis of trials which randomized patients with CRAO to intravenous thrombolysis. There is currently no approved acute revascularization treatment for CRAO and off-label treatments in everyday clinical practice are widely used with great uncertainty. Although there is a need for robust high-quality evidence for supporting such treatment decisions, patient recruitment in trials of acute treatments in CRAO is significantly limited because its occurrence is not as frequent as acute ischemic stroke and because of hospital admission delays. It is unlikely that any single RCT will be powered to demonstrate statistically significant but relevant visual functional improvement. Here lies the importance of this IPDMA, which will probably influence treatment guidelines worldwide. I congratulate the authors for this cooperation effort.

I have only minor comments:

1. Studies performing MRI in patients with CRAO show a relatively high proportion of patients with silent acute ischemic lesions, but concurrent ischemic stroke (i.e. clinical neurological symptoms associated with acute ischemic lesions) is rare. This should be corrected in the introduction.

The authors greatly appreciate this comment. We have revised the introduction section accordingly.

2. After the recent publication of THEIA, the phrase “with no published cases when treatment was given within 4.5 hours of time last known well” is no longer true. Please correct.

Thank you for this comment. We respectfully adopt a different perspective, the ICH reported in THEIA was asymptomatic and thus we would propose not to include this particular event. However, there was a sICH causing death in the TenCRAOS trial that was presented at ESOC in May. We have updated the manuscript to reflect this conference result.

3. Under the paragraph “Rational for individual participant data meta-analysis” please update the part referring to THEIA, which is now published.

Thank you. The authors apologize for this omission. We have updated this section.

4. Please use another term (or sentence) for describing “grey literature”, for readability.

Thank you, we have revised and updated this sentence to improve readability.

5. Please specific in which time window will sICH be defined (also according to SITS-MOST?).

Thank you very much. We have updated this definition to clarify that we will use the SITS-MOST definition and to clarify that it will be within 36 hours of time last known well.

6. The different trials which will be included in the IPMA appear to have defined the primary endpoint differently. I think it would be useful for the readers that the authors explain why they chose logMAR 0.50 as the primary endpoint, and what it means in terms of visual function (it it described under the section “Secondary Endpoints” but it should be presented under the section “Primary Endpoint”).

Thank you. We have corrected this section as follows:

“The primary end point will be attainment of a final BCVA equal to or better than 20/63 (logMAR of ≤0.5). This threshold was chosen because i) it indicates a patient-centered degree of visual recovery (classed at the lower threshold of “mild visual impairment” by the WHO), ii) because the requirement for this degree of recovery introduces a conservative bias in to the analysis, and iii) because an apparent improvement to 20/63 cannot be achieved via eccentric fixation.”

7. Please correct “European Stroke Organization annual conference” to European Stroke Organization Conference”.

Thank you, we apologize and have made this correction.

● Reviewer #2:

This registered report protocol is well written, timely, and methodologically robust. The clinical question is highly relevant, the rationale for an IPD meta-analysis is clearly articulated, and the planned analyses are detailed, appropriate, and transparent. I support publication after addressing the following minor points for clarification and completeness:

1. Justification of the 0–4.5 hour time window

You have chosen a ≤4.5 h time frame from last known well (LKW) to IV thrombolysis (IVT), aligned with cerebral ischemic stroke practice and with much of the CRAO thrombolysis literature. However, in CRAO there is an ongoing debate about the role of collateral flow and retinal “penumbra” and whether some patients may retain salvageable tissue beyond 4.5 hours.

• Please expand the rationale for choosing 4.5 hours rather than including trials with treatment up to 24 hours (or other extended windows).

• It would be helpful to explicitly relate this to:

o The non-human primate data suggesting variable retinal tolerance (e.g., the 105 vs 240 minute survival times and how these were interpreted).

o The hypothesis that collateral choroidal flow and individual vascular risk profiles might modulate the true therapeutic window.

o The practical/clinical trade-off between maximizing internal validity (homogeneous early time window) vs external validity (relevance to real-world late presenters).

Even if you ultimately keep the 0–4.5 h window as an inclusion criterion, a more explicit discussion in the Methods and/or Discussion of why late-presenting patients (e.g., up to 24 h) and extended windows are not incorporated would strengthen the protocol and preempt future interpretative debates.

Thank you very much. The authors greatly appreciate this comment. We have updated the discussion section as follows:

“Fourth, we have chosen to focus on the 4.5 hour window from time last known well to treatment. The rationale for this choice is i) it is our anticipation that all trials examined as part of our systematic review will adopt this threshold and ii) these results will be of the most immediate translational relevance. However, the results obtained cannot be generalized to those patients who, in future, might be considered for treatment beyond 4.5 hours of time last known well based on estimates of potential efficacy using markers of retinal structure and function that could be used to infer retinal viability in select patients.”

2. More granular onset-to-treatment time subgroup analyses

You already plan to model time-to-treatment and its association with visual outcomes. Given the central importance of timing in CRAO, I would encourage you to pre-specify more granular time windows as part of your subgroup or sensitivity analyses.

For example, you could consider:

• 0–1.5 hours

• 1.5–3 hours

• 3–4.5 hours

• (If data permit, an exploratory >4.5 h subgroup where such patients exist in any contributing trial.)

This does not have to change your primary analysis, which can remain continuous (or modeled with splines), but explicitly listing these time strata as planned exploratory subgroups would:

• Enhance interpretability for clinicians (who often think in discrete time windows).

• Potentially identify clinically meaningful thresholds within the 0–4.5 h window.

• Make it easier to align your findings with how stroke systems of care are organized in practice.

Thank you very much for this comment. We have updated our section on subgroup analyses to include this analysis which we feel will greatly enrich this meta-analysis.

3. Safety endpoints – explicit inclusion of angioedema

In the Methods you already plan to collect several safety endpoints, including symptomatic ICH and systemic hemorrhage. You also mention angioedema later in the text, but it would be helpful to:

• Explicitly list “orolingual angioedema” among the predefined safety endpoints in the main safety endpoint section, alongside sICH and systemic hemorrhage.

• Clarify whether angioedema will be:

o Analyzed descriptively only (expected low incidence), or

o Included in any pooled or exploratory safety models, if numbers permit.

Given the known association of IVT with orolingual angioedema, this explicit inclusion will be reassuring both for readers and guideline developers focusing on safety.

Thank you very much. We have explicitly added orolingual angioedema to the list of end points, the statistical ana

---

## [Decision Letter · Decision Letter 1]

29 Jan 2026

Intravenous Thrombolysis for Acute Central Retinal Artery Occlusion: Protocol For a Systematic Review and Individual Participant Data Meta-Analysis of Randomized Controlled Trials

PONE-D-25-53172R1

Dear Dr. Mac Grory,

We’re pleased to inform you that your manuscript has been judged scientifically suitable for publication and will be formally accepted for publication once it meets all outstanding technical requirements.

Kind regards,

Ogugua Ndubuisi Okonkwo, M.D.

Academic Editor

PLOS One

Additional Editor Comments (optional):

Reviewers' comments:

Reviewer's Responses to Questions

**Comments to the Author**

1. Does the manuscript provide a valid rationale for the proposed study, with clearly identified and justified research questions?

Reviewer #1: Yes

Reviewer #2: Yes

2. Is the protocol technically sound and planned in a manner that will lead to a meaningful outcome and allow testing the stated hypotheses?

Reviewer #1: Yes

Reviewer #2: Yes

3. Is the methodology feasible and described in sufficient detail to allow the work to be replicable?

Reviewer #1: Yes

Reviewer #2: Yes

4. Have the authors described where all data underlying the findings will be made available when the study is complete?

Reviewer #1: Yes

Reviewer #2: Yes

5. Is the manuscript presented in an intelligible fashion and written in standard English?

*PLOS ONE*

Reviewer #1: Yes

Reviewer #2: Yes

You may also provide optional suggestions and comments to authors that they might find helpful in planning their study.

Reviewer #1: I have no further comments. All questions were addressed by the authors and the manuscript was changed accordingly.

Reviewer #2: The authors have adequately addressed all previous comments, and the revised manuscript is substantially improved. Congratulations on this important effort. I eagerly anticipate the final results of the IPD analysis, which will be of great importance to the field.

**Do you want your identity to be public for this peer review?** For information about this choice, including consent withdrawal, please see our Privacy Policy

Reviewer #1: No

Reviewer #2: **Yes:** Klearchos Psychogios

---

## [Editor Report · Acceptance letter]

PONE-D-25-53172R1

PLOS One

Dear Dr. Mac Grory,

I'm pleased to inform you that your manuscript has been deemed suitable for publication in PLOS One. Congratulations! Your manuscript is now being handed over to our production team.

Kind regards,

on behalf of

Prof Ogugua Ndubuisi Okonkwo

Academic Editor

PLOS One